# Invasive *Aspergillosis* in the Intensive Care Unit

**DOI:** 10.3390/diagnostics12112712

**Published:** 2022-11-06

**Authors:** Liam Townsend, Ignacio Martin-Loeches

**Affiliations:** 1Department of Intensive Care Medicine, Multidisciplinary Intensive Care Research Organization (MICRO), St. James’s Hospital, D08 NHY1 Dublin, Ireland; 2Department of Clinical Medicine, School of Medicine, Trinity College Dublin, D02 PN91 Dublin, Ireland; 3Hospital Clinic, Institut D’Investigacions Biomediques August Pi i Sunyer (IDIBAPS), Universidad de Barcelona, Ciberes, 08036 Barcelona, Spain

**Keywords:** *Aspergillus*, ICU, diagnostics, galactomannan

## Abstract

Invasive pulmonary aspergillosis (IPA) is a serious condition resulting in significant mortality and morbidity among patients in intensive care units (ICUs). There is a growing number of at-risk patients for this condition with the increasing use of immunosuppressive therapies. The diagnosis of IPA can be difficult in ICUs, and relies on integration of clinical, radiological, and microbiological features. In this review, we discuss patient populations at risk for IPA, as well as the diagnostic criteria employed. We review the fungal biomarkers used, as well as the challenges in distinguishing colonization with *Aspergillus* from invasive disease. We also address the growing concern of multidrug-resistant *Aspergillosis* and review the new and novel therapeutics which are in development to combat this.

## 1. Introduction

*Aspergillus* spp. are a family of ubiquitous fungi. However, they have the potential to cause a wide spectrum of diseases in susceptible hosts. This disease spectrum ranges from reactive allergic symptoms to invasive pulmonary disease. Host immunity typically prevents *Aspergillus* from causing disease. However, alterations in host immune interactions with *Aspergillus* leads to the development of pathologies. On one end of the spectrum, atopic patients can develop an allergic-type response with an over-exuberant immune response, leading to the development of conditions such as allergic bronchopulmonary aspergillosis. Patients with preexisting lung structural lung disease, in particular chronic obstructive pulmonary disease (COPD) and cystic fibrosis, can be colonized by *Aspergillus* and develop chronic bronchitis as a result. This is not a risk factor unique to aspergillosis, as patients with structural lung disease are at increased risk of infection from a variety of pathogens [1]. At the opposite end of the spectrum, immunocompromised hosts are at risk of developing invasive disease, with invasive pulmonary aspergillosis (IPA) being the commonest mold infection in immunocompromised hosts [2]. Invasive fungal infections, in particular IPA, are a significant contributor to morbidity and mortality among intensive care unit (ICU) patients [3], as well as a significant economic burden [4]. Given the difficulty in diagnosis of IPA, it is difficult to accurately assess the number of deaths attributed to it. However, the incidence of serious fungal infections in critically unwell patients has increased in recent years and carries a high mortality rates [5,6]. There are approximately 0.2 million cases of IPA each year, although this is likely an underestimate [3]. The 90-day mortality in patients with hematological malignancy who develop IPA reaches 80%, while IPA in non-immunosuppressed ICU patients has been reported to occur in up to 15% of cases. Despite the clinical impact of IPA, early diagnosis and prompt initiation of therapy is often delayed [7]. The identification of at-risk patients, use of appropriate diagnostics, and early initiation of treatment for IPA are essential in the care of ICU patients. Intensive care physicians should be aware of the risk factors for the development of IPA and the clinical features suggestive of the condition, as well as the diagnostic investigations and criteria and emergent therapies.

## 2. Risk Factors

Inhaled *Aspergillus* conidia are usually cleared by host innate immune defenses at the respiratory epithelium prior to hyphae formation, typically by resident alveolar macrophages, and do not cause disease [8]. However, significant invasive disease can occur in susceptible hosts, with conidia germinating and transforming into hyphae. Recognizing at-risk patients for IPA prior to the development of invasive disease is essential for early detection. Prolonged neutropenia and post-transplant immunosuppression are the most widely recognized risk factors for the development of IPA. Both the duration and severity of neutropenia are associated with increased risk of IPA [9]. Allogeneic hematopoietic stem cell transplant recipients have several risk factors for the development of IPA. They are profoundly neutropenic following their transplant conditioning regime, they may develop acute graft-versus-host disease post-transplant, and may also develop chronic graft-versus-host disease [10]. Recent evolution in methods used in hematopoietic stem cell transplantation, with less toxic chemotherapy and reduction in the duration of neutropenia, have been accompanied by a reduction in IPA mortality [11]. The epidemiology and disease pattern of IPA in the setting of hematologic and oncologic conditions have been described well previously [12,13,14,15].

However, there is growing recognition of other risk factors associated with IPA in critical care. Factors associated with IPA mortality include degree of neutropenia and concomitant end-organ failure, indicating that a more severely ill and morbid patient is at increased risk of death [16]. Preexisting end-organ diseases, in particular chronic obstructive pulmonary disease (COPD) and cirrhotic liver disease, are associated with increased IPA risk [17,18,19]. The use of high-dose corticosteroids is also associated with the development of IPA [20]. Importantly, use of corticosteroids prior to hospital admission has been demonstrated to increase IPA risk; IPA risk has also been increased among those who receive corticosteroids during their admission. This latter at-risk population is likely to grow, given the use of corticosteroids in patients with septic shock requiring vasopressor therapy in ICU settings [21,22]. Other iatrogenic risk factors include the use of T-lymphocyte immunosuppressants such as calcineurin inhibitors and TNFα inhibitors, as well as B-cell immunosuppressants such as ibrutinib. The combination of immunosuppression and altered respiratory anatomy is seen in lung transplant recipients, who as a result have a higher incidence of IPA than other solid-organ transplant patients [23]. This may in part be due to single-lung transplant recipients already being colonized by *Aspergillus* [24].

A small but important at-risk group are patients with inherited immunodeficiencies resulting in quantitative or qualitative lymphocyte deficits. IPA is a significant cause of death among patients with chronic granulomatous disease, whose phagocytic cells are unable to complete the oxidative burst due to NADPH oxidase dysfunction. This leads to an over-active inflammatory response and disruption of homeostatic and anti-inflammatory mechanisms, resulting in chronic wound formation and endothelial disruption. The mortality rate associated with IPA may reflect underlying disease severity in this cohort. Other less common primary immunodeficiencies that are associated with IPA include hyper IgE syndrome with recurrent infections, primary T-cell deficiencies, and mitochondrial disorders [25].

## 3. Pathogenesis

The presence of *Aspergillus* spp. in respiratory samples may represent colonization rather than IPA [26,27,28]. The host immune response is responsible for preventing clinical disease in this context. The innate immune response at the respiratory epithelium prevents the development of severe invasive disease. *Aspergillus*-specific Toll-like receptors coordinate the local immune response, ensuring homeostasis between pro- and anti-inflammatory activities and the continuity of the endothelium [29]. This immune homeostasis is further regulated by adaptive immune responses, with specific Th1 and Th2 cell responses to *Aspergillus* [30]. However, in a susceptible host with *Aspergillus* colonization, this may progress to invasive disease [31]. In addition to hosts with the chronic conditions and comorbidities already described, severe intercurrent illness is associated with the development of IPA. Indeed, admission to ICUs is an independent IPA risk factor [17], with patients admitted to ICUs with respiratory tract infections having a significantly increased risk [32].

The increased incidence of IPA in previously well, critically ill patients is most likely as a result of a combination of respiratory epithelium disruption and aberrant host immune responses [33]. The inflammatory homeostasis at the alveolus is disrupted during viral infection, with a large amount of proinflammatory cytokines produced, leading to local tissue disruption. This results in the development of acute respiratory distress syndrome (ARDS), with associated architectural disruption and immune disturbance at the alveolus. These changes are demonstrated in Figure 1.

The progression of *Aspergillus* from respiratory colonization to invasive infection is characterized by angioinvasion of the vascular endothelium [34]. This epithelial disruption allows for the translocation of *Aspergillus* into the surrounding tissues and bloodstream, resulting in local thrombosis and infarction [35]. This can cause sequestration of *Aspergillus*-infected tissue. These pockets of infected tissue have impaired vascular supply, and it is challenging to deliver adequate levels of antifungal therapy to them, often resulting in treatment failure [36]. *Aspergillus* propagates the effect of local ischemia and thrombosis by directly inhibiting angiogenesis [37]. Angioinvasion is most commonly seen in neutropenic patients, whereas it may develop later in non-neutropenic hosts, leading to delays in diagnoses. This diagnostic challenge is reflected by IPA being the most commonly missed diagnosis of infection among post-mortem findings for ICU patients [38].

Given that IPA occurs at higher frequencies in patients with structural lung disease, it is not surprising that it is well recognized as a complication of severe influenza A infection, resulting in influenza-associated pulmonary aspergillosis (IAPA) [39,40,41]. IAPA can be seen in patients who do not have traditional risk factors for IPA. It is thought that influenza has the ability to induce immune paralysis in these patients [42]. Influenza also disrupts alveolar epithelial cell junctions, impairing their structural integrity in a manner similar to that shown in Figure 1 [43]. The development of IAPA is associated with significantly higher mortality [44]. The SARS-CoV-2 pandemic raised concerns that similar rates of IPA would be seen in critical-care COVID-19 patients, with the development of so-called COVID-associated pulmonary aspergillosis (CAPA). These concerns were based on evidence that showed that SARS-CoV-2 is associated with endothelial dysfunction [45,46], while severe COVID-19 is associated with severe lymphopenia, potentially impairing the adaptive response to *Aspergillus* [47,48]. CAPA, similarly to IAPA, has been found among patients who do not have traditional risk factors for IPA [49]. However, there have been significant variations in the reported incidence of IPA in these patients, with levels not approaching those seen in influenza [49,50,51,52,53,54]. Nevertheless, the high mortality of CAPA has necessitated the development of specific criteria for its diagnosis.

## 4. Clinical Presentation

Invasive aspergillosis typically begins in the lungs and sinuses, given that acquisition occurs via inhalation. *Aspergillus* spp. within the bronchioles and alveoli has the ability to bind surfactant proteins and transit across the alveolar epithelium and into the bloodstream, resulting in IPA. The clinical presentation of IPA is protean, and there is a wide spectrum of illnesses within it [55]. The development of pulmonary infarction is associated with a classic triad of pleuritic chest pain, fever, and hemoptysis, but this is nonspecific and is not seen in all cases. Invasive rhinosinusitis is associated with facial/ocular pain and nasal congestion, while disseminated disease can present with features of meningism, endophthalmitis, or endocarditis [56]. Involvement of the large airways can result in tracheobronchitis. These patients typically have profound shortness of breath and productive cough. They may expectorate mucus plugs. There is a spectrum of *Aspergillus* tracheobronchitis, from including ulcerative, obstructive, and pseudomembranous. Sinus involvement can mimic disease caused by mucormycosis, with facial and peri-ocular pain as well as fever and nasal congestion. Severe invasive sinus disease can result in CNS involvement, with venous sinus thrombosis. The lack of specificity of these symptoms is compounded in the ICU setting, where patients may be intubated and are often unable to describe their symptoms, as well as limiting the scope for physical examination. The clinical presentation can be further distorted in patients who have a preexisting respiratory infection, as is the case with CAPA and IAPA. This places a large emphasis on radiological and mycological findings in the diagnosis of IPA. This is exacerbated by the low rate of *Aspergillus*-positive blood cultures, even in the context of angioinvasion.

## 5. Diagnosis

The diagnosis of IPA can be challenging. Tissue diagnosis remains the gold standard, with culture (either of tissue sampling or blood) or polymerase chain reaction (PCR) of molds from tissue samples. Histological features are nonspecific, with PCR having increased sensitivity in comparison with histology or fungal culture [57]. There is often pyogranulomatous inflammation and inflammatory necrosis with histology, while neutropenic hosts may develop angioinvasion with hemorrhagic necrosis [58]. These findings are not specific to IPA, and require the identification of fungal hyphae to confirm the diagnosis. PCR techniques are becoming more widely available with improved reproducibility [59]. However, the European *Aspergillus* PCR Initiative recommends two positive specimens to conclusively make a diagnosis [60].

Measurement of galactomannan, an *Aspergillus*-specific antigen, can be used to support a diagnosis of IPA [61]. Galactomannan (GM), a β-D glucan, is a major component of the *Aspergillus* cell wall. It is preferred to measuring total β-D glucan for IPA, given its increased specificity. Angioinvasion by *Aspergillus* leads to galactomannan release into the circulation. It can be measured in serum, plasma, BAL samples, or CSF. Respiratory samples are considered superior to blood tests, particularly in non-neutropenic patients [62]. Galactomannan results are reported as optical densities, providing a galactomannan index (GMI). Assessment of galactomannan can support or rule out IPA, with a GMI > 0.8 associated with invasive disease and a GMI < 0.5 being associated with a very low likelihood of IPA in neutropenic patients [63,64]. A serum GMI between 0.5 and 0.8 is equivocal and must be interpreted in the context of other clinical and radiological features. GMI value cutoffs from other sites remain controversial and must be interpreted within the clinical context, with a higher GMI required for patients known to be colonized with *Aspergillus*, such as those with COPD [65]. The performance and reliability of galactomannan assays can vary based on clinical situations [66]. GM is cleared by neutrophils; therefore, its utility in IPA diagnoses among non-neutropenic patients is limited [67]. BAL GM levels and sequential serum GM levels can increase the sensitivity and specificity of testing among non-neutropenic patients [62,65,68] and are also superior to serum levels in patients admitted with concomitant respiratory tract infections [69]. False-positive results have been well-described when patients are receiving concurrent beta-lactam antibiotics such as piperacillin-tazobactam, and are seen in both serum and BAL measurements of galactomannan [70,71,72]. False-positive GM may also be seen in infection with other fungi with cell walls similar to *Aspergillus* galactomannan, such as *Histoplasma capsulatum* [73]. This is mainly of clinical relevance in areas in which *Histoplasma* is endemic. Interestingly, GMI can be prognostic as well as diagnostic, with a GMI >2 associated with poor prognosis [74].

The radiological features seen in IPA are varied and nonspecific. The plain film of the chest lacks the sensitivity to detect IPA, and as such all patients with suspected IPA should undergo computed tomography (CT) imaging [75]. The characteristic CT features of IPA include peri-bronchial consolidation, ground-glass opacities, and the halo sign, which is a nodule with surrounding ground-glass opacities, representing local hemorrhage at the site of fungal infection [76,77]. Other angio-invasive pathogens such as *Pseudomonas* are also capable of producing halo signs. An air-crescent sign may be seen and is a consequence of pulmonary necrosis [78]. The frequency of these findings varies depending on the underlying condition of the patient, with neutropenic patients more likely to show halo signs [79], while patients with solid-organ transplants are more likely to have consolidation or solid masses visible on CT [80].

Due to the complexity of recognizing and diagnosing IPA, several diagnostic algorithms for have been developed based on expert-consensus-based discussions. These include the European Organization for Research and Treatment of Cancer and the Mycoses Study Group (EORTC/MSG) [81], and the biomarkers for *Aspergillus* in the ICU (BM-AspICU) [82], with additional modifications for the diagnosis of IAPA and CAPA [83]. These algorithms describe the criteria needed for proven IPA, as well as allowing for probable and possible cases. Proven IPA requires the presence of positive culture or PCR of *Aspergillus.* Probable IPA requires a susceptible host with characteristic radiological features in addition to supporting mycological evidence. As previously described, susceptible hosts include patients with neutropenia, hematologic or oncologic malignancies, and those receiving high-dose corticosteroids. The mycological evidence is provided by the isolation of *Aspergillus* species and the measurement of galactomannan. These criteria are summarized in Table 1.

The emergence of IAPA and CAPA lead to concerns that not all cases of IPA were being captured using the EORTC/MSG definition [81]. In particular, CAPA patients may not have any traditional host risk factors. They also may not develop characteristic CT changes, and any CT changes that do occur may overlap with those seen in severe SARS-CoV-2 infection. In order to address this, separate criteria have been developed for the diagnosis of CAPA. The European Confederation of Medical Mycology (ECMM) and the International Society for Human and Animal Mycology (ISHAM) have developed CAPA diagnostic criteria. They suggest that CAPA should be considered in any patient who has recrudescence of fever after being fever-free for more than 72 h, or any patient with worsening respiratory status or new respiratory symptoms. Investigations are similar to those used in non-COVID-19 IPA, with CT imaging and respiratory tract sampling for fungal culture, PCR, and galactomannan assay. CT imaging in severe COVID-19 may be difficult to distinguish from IPA. However, patients with cavitating lung lesions or multiple pulmonary nodules should undergo extensive IPA investigation, as these are atypical for COVID-19 pneumonitis [49]. A single positive BAL sample in the context of characteristic imaging is highly suggestive of IPA. Performing BAL in patients with COVID-19 pneumonitis is not routinely recommended due to risks to the healthcare worker performing the procedure [84]. Therefore, circulating biomarkers are often used to aid diagnosis. Serum galactomannan assay is highly specific in patients with CAPA, as these are non-neutropenic, although the sensitivity is low. As such, serum galactomannan has limited utility in excluding CAPA. ECMM/ISHAM recommend that screening with serum galactomannan should be performed three times per week in all patients that have a positive SARS-CoV-2 PCR [85]. Proven CAPA is considered if there is definitive histological or microscopic evidence of *Aspergillus* invasion of tissue, or positive *Aspergillus* PCR from a sterile site. A diagnosis of probable CAPA is made if characteristic CT finding are present in addition to mycological evidence, with either positive PCR from BAL aspirate or positive galactomannan. Possible CAPA is diagnosed based on characteristic CT findings as well as supporting mycological evidence from a site other than BAL.

## 6. Treatment

While an exhaustive description of treatment options for IPA is beyond the scope of this review, it is important to address emerging treatment options, particularly in light of growing antifungal resistance. Host-directed therapies, such as those targeting the pulmonary endothelial disruption associated with IPA, have yet to demonstrate clinical utility. Nevertheless, as stem cell therapies are developed, their application in the restoration of endothelial dysfunction and homeostasis may become important additions to antifungal therapies [86,87]. Antifungal therapies have generally been drawn from four classes of drugs (azoles, echinocandins, polyenes, and 5-flucytosine). First-line empiric therapy for IPA is with azoles, with voriconazole the preferred agent [88]. Voriconazole therapy can be challenging, as it requires therapeutic drug monitoring. Isuvaconazole has become an attractive alternative to voriconazole. Its performance is noninferior in clinical trials, and it has a favorable side-effect profile [89]. Resistance to all antifungal classes has been identified [90], and there is growing levels of *Aspergillus* resistance to voriconazole [91]. Fungi demonstrate an ability to develop resistance via multiple mechanisms, including drug efflux pumps and genetic alteration of drug targets [92]. Furthermore, fungal infections have been shown to persist even in the presence of antifungal agents that are shown to be sensitive in culture. This is due to selective survival pressure driving fungal resistance in vivo, a concept known as drug tolerance [93]. The widespread use of antifungal therapies in agriculture and industry has been directly linked with human infection of drug-resistant *Aspergillus* [94]. Triazole-resistant *Aspergillus* infection has been seen with increasing frequency in patients without prior antifungal therapy, as a result of azole use outside the clinical setting [95,96]. The emergence of resistance, coupled with the inability of infected hosts to clear *Aspergillus* infection due to underlying immune defects or comorbidities, is a significant clinical problem [97]. The utility of PCR as a diagnostic tool, in preference to biomarkers or direct tissue microscopy, allows the identification of specific *Aspergillus* species, as well as potentially identifying genes associated with resistance [98]. In order to prevent the onward propagation of antifungal resistance, the antimicrobial stewardship principles of short-duration, high-concentration therapies at need to be applied. Alternative methods of drug delivery such as nebulization have been trialed, as well as the use of therapeutic drug monitoring [99]. Antifungal susceptibility testing is often not routinely performed. Susceptibility testing should be carried out if the patient has received extensive antifungal treatments in the past, is in an area with known high levels of azole resistance, or is failing to respond to appropriate empiric therapy. It is recommended that a minimum of five colonies should be tested for resistance, given that azole-sensitive and azole-resistant species may occur simultaneously [100].

Novel drug classes have been developed: fosmanogepix, a Gwt1 inhibitor; ibrexafungerp, a triterpenoid; and olorofim, a dihydroorotate dehydrogenase inhibitor. There are also innovations in the delivery of preexisting drug classes, with the emergence of the nebulized triazole opelconazole. These new agents are summarized in Table 2.

Fosmanogepix is a pro-drug of manogepix and acts by preventing the maturation of mannoproteins [101]. These mannoproteins are essential components of the cell wall. Interfering with their function has multiple downstream effects, with reduction in biofilm formation, endoplasmic reticulum stress, and impaired cell wall integrity [102]. Fosmanogepix has activity against a broad array of resistant fungi due to its novel mechanism of action, and it has been shown to be effective in vitro against triazole-resistant *Aspergillus*, with clinical trials in process [103,104]. It has high oral bioavailability, making it an attractive option for IPA cases where intravenous liposomal amphotericin B is the only therapeutic option [105]. It has also been shown to have a favorable safety profile, with no evidence of nephrotoxicity [106]. Fosmanogepix is already under consideration by the United States Food and Drug Administration.

Ibrexafungerp inhibits 1, 3-beta-D-glucan in fungal cell walls of *Aspergillus* and other fungi, preventing its synthesis in a manner similar to echinocandins. However, the site of action is distinct to echinocandins, resulting in minimal cross-resistance [107]. Similar to fosmanogepix, ibrexafungerp is also available as an oral formulation, and shows in vitro ability to inhibit the growth of *Aspergillus*. It is metabolized via the cytochrome P450 pathway, but modulators of this pathway have a less marked effect on ibrexafungerp levels than on preexisting azoles. As such, it has been shown to be tolerated well with minimal side effects [108]. It has already been approved for treatment of *Candidiasis*, and is undergoing review for *Aspergillosis* [109]. The final novel drug in development, olorofim, inhibits pyrimidine synthesis by fungi, inhibiting their growth. It shows activity against triazole-resistant *Aspergillus* [110]. A potential pitfall to its use is that it is metabolized via the cytochrome p450 system, and as such has several clinically significant drug–drug interactions [111]. Olorofim remains under clinical investigation and has not yet been licensed for use, although there are several Phase IIb clinical trials underway.

Perhaps the most exciting of the new therapeutic agents in development comes from a preexisting drug class. Opelconazole is a novel triazole, acting in a similar manner to preexisting triazoles. It inhibits lanosterol 14α-demethylase, leading to fungal membrane dysfunction [112]. Importantly, it has been developed with nebulization as the preferred route of administration. Early pharmacokinetic data show that it reaches excellent concentrations within the lungs, and has minimal systemic absorption which reduces toxicity [113]. It has been shown to be highly effective against multiple fungal species, including *Aspergillus*, and shows superior activity to other azoles [112]. Opelconazole is an attractive option in patients who are at risk of IPA, including the growing cohort of patients being treated with immunosuppressive agents, as well as those with pulmonary disease, and patients with COPD, post-lobectomy, and post-transplant conditions [114]. Furthermore, its reassuring safety profile with low systemic absorption makes it a good candidate for combination therapy with the other novel agents being developed [115].

## 7. Conclusions

*Aspergillosis* and IPA remain significant clinical challenges. There is a growing cohort of patients with risk factors for IPA development, including non-neutropenic patients. Furthermore, advances in critical care mean that patients may develop IPA in the absence of traditional risk factors, as a consequence of their acute severe illness. The diagnosis of IPA requires the treating clinician to evaluate host factors, clinical features, radiological characteristics, and appropriate biomarkers from relevant clinical sites. They also must be conscious of the diagnostic criteria, and the variations that exist between criteria across different underlying conditions. IPA treatment can be challenging, exacerbated by the development of antifungal resistance. The advent of new therapeutic agents, coupled with increased awareness of IPA risks and improved access to diagnostics, should allow intensivists to promptly recognize, diagnose, and treat this condition.

## Figures and Tables

**Figure 1 diagnostics-12-02712-f001:**
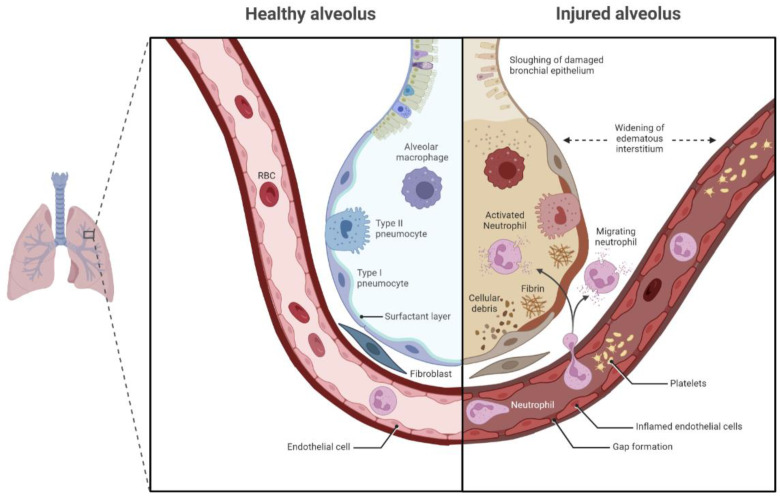
**Healthy alveolus and injured alveolus in acute respiratory distress syndrome.** Healthy alveolus with resident immune cells on the **left**, with injured alveolus in ARDS on the **right**, showing alterations in immune cell populations and endothelial dysfunction. Image created at Biorender.com (accessed on 1 October 2022).

**Table 1 diagnostics-12-02712-t001:** European Organization for Research and Treatment of Cancer and the Mycoses Study Group Diagnostic Criteria for IPA.

○ **Proven invasive pulmonary aspergillosis**
Histopathological evidence of *Aspergillus* hyphae with tissue damage; Blood culture positive for *Aspergillus.*
○ **Probable invasive pulmonary aspergillosis (at least one from each category is needed)**
1. Host factors
Prolonged neutropenia; stem cell transplant recipient; T-cell immunosuppressant; inherited immunodeficiency.
2. Clinical features on imaging
Well-circumscribed dense lesion; cavity; air-crescent sign.
3. Mycological criteria
Cytology, microscopy, or culture on sputum, BAL, or bronchial brushing indicating *Aspergillus* elements or positive *Aspergillus* culture; positive galactomannan in blood or BAL samples.
○ **Possible invasive pulmonary aspergillosis**
Presence of both host factors and clinical features of probable invasive pulmonary aspergillosis, but not meeting mycological criteria.

**Table 2 diagnostics-12-02712-t002:** Novel antifungal therapies for invasive pulmonary aspergillosis.

Drug	Class	Mechanism of Action	Route of Administration
Foxmanogepix	Gwt1 inhibitor	Inhibit mannoprotein maturation and impair fungal cell wall integrity	Oral
Ibrexafungerp	Triterpenoid	Inhibit β-D glucan synthesis and impair fungal cell wall integrity	Oral
Olorofim	Dihydroorotate dehydrogenase inhibitor	Inhibit pyrimidine synthesis	Oral
Opelconazole	Triazole	Lanosterol 14α-demethylase inhibition	Nebulized

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
