# Peer review of "Invasive Aspergillosis in the Intensive Care Unit"

_diagnostics, 2022, doi:10.3390/diagnostics12112712_

Round 1

Reviewer 1 Report

Your comprehensive and up-to-date review is very well written and deserves publication. I have no major comments.

Minor comments:

1)    You cited the reference stating that galactomannan (GM) from respiratory samples is considered superior to blood test in non-neutropenic patients (page 7, line 163), but quite often laboratories use assays validated for serum (cut-offs only for serum are provided). Could you comment on this subject and provide some advice how use these assays for respiratory samples? Could you signal in this part of the manuscript, that in COVID-19 serum samples are recommended, as you have written on page 11, lines 219-222? 

2)    The subheading “Conclusions” is lacking on page 15, between the lines 291 and 292.

Reviewer 2 Report

The manuscript described invasive pulmonary aspergillosis (IPA). The diagnosis of IPA can be difficult in the intensive care unit (ICU). The authors showed the feature of IPA. The diagnosis of IPA and chemotherapy using foxmanogepix, ibrexafungerp, olorofim, or opelconazole were suggested. Thus, these findings will be useful for IPA. Therefore, the manuscript is not too excellent to be published after revision. In other words, the manuscript is so excellent that it should be published after revision.

Comments

(1) How many people died of IPA per year?

(2) Does chronic obstructive pulmonary disease (COPD) occur concurrently with infection of other germs than Aspergillus species? Influenza?

(3) Epithelial disruption allows for the translocation of Aspergillus into the surrounding tissues and bloodstream. When epithelial disruption is restored, will disease derived from IPA be cured?

(4) Are respiratory samples considered superior to blood tests, with respect to accuracy and/or simplicity?

(5) Did chemotherapy using foxmanogepix, ibrexafungerp, olorofim, or opelconazole exhibit good results for IPA without side effects? Will they be available clinically?

That is all.
